# Multi-Stress Induction of the *Mycobacterium tuberculosis* MbcTA Bactericidal Toxin-Antitoxin System

**DOI:** 10.3390/toxins12050329

**Published:** 2020-05-16

**Authors:** Kanchiyaphat Ariyachaokun, Anna D. Grabowska, Claude Gutierrez, Olivier Neyrolles

**Affiliations:** 1Institut de Pharmacologie et de Biologie Structurale, IPBS, Université de Toulouse, CNRS, UPS, 205 route de Narbonne, 31400 Toulouse, France; kanchiyaphat.a@ubu.ac.th (K.A.); dr.anna.grabowska@gmail.com (A.D.G.); olivier.neyrolles@ipbs.fr (O.N.); 2Department of Biological Science, Ubon Ratchathani University, 85 Sathollmark Rd., Warin Chamrab, Ubon Ratchathani 34190, Thailand; 3Department of Biophysics and Human Physiology, Medical University of Warsaw, 02-091 Warsaw, Poland

**Keywords:** tuberculosis, toxin-antitoxin systems, bacterial cell death, NAD^+^, stress-response

## Abstract

MbcTA is a type II toxin/antitoxin (TA) system of *Mycobacterium tuberculosis*. The MbcT toxin triggers mycobacterial cell death in vitro and in vivo through the phosphorolysis of the essential metabolite NAD^+^ and its bactericidal activity is neutralized by physical interaction with its cognate antitoxin MbcA. Therefore, the MbcTA system appears as a promising target for the development of novel therapies against tuberculosis, through the identification of compounds able to antagonize or destabilize the MbcA antitoxin. Here, the expression of the *mbcAT* operon and its regulation were investigated. A dual fluorescent reporter system was developed, based on an integrative mycobacterial plasmid that encodes a constitutively expressed reporter, serving as an internal standard for monitoring mycobacterial gene expression, and an additional reporter, dependent on the promoter under investigation. This system was used both in *M. tuberculosis* and in the fast growing model species *Mycobacterium smegmatis* to: (i) assess the autoregulation of *mbcAT*; (ii) perform a genetic dissection of the *mbcA* promoter/operator region; and (iii) explore the regulation of *mbcAT* transcription from the *mbcA* promoter (P*_mbcA_*) in a variety of stress conditions, including in vivo in mice and in macrophages.

## 1. Introduction

*Mycobacterium tuberculosis* (*Mtb*), the etiological agent of tuberculosis (TB), is a strictly human-adapted obligate pathogen of major public health importance (WHO, Global Tuberculosis report, 2019). Indeed, TB was responsible for 1.5 million deaths and about ten million new TB cases were reported in 2018. Of particular concern, an estimated 484,000 new infections were due to *Mtb* strains resistant to one or more frontline antibiotics in 2018, and 80% of these *Mtb* strains were multi-drug resistant (MDR). Therefore, the development of novel therapies against TB is urgently needed. In line with this, we recently started to explore the potential utilization of intrinsic bactericidal toxin/antitoxin (TA) systems as new therapeutic tools [1]. TA systems are genetic modules encoding an endogenous protein toxin, which targets an essential process in the bacterial cell, and a toxin-neutralizing antidote, the so-called antitoxin [2,3,4]. Under favorable conditions, the antitoxin is stable and prevents auto-intoxication of the TA-producing cell. Although the precise mechanisms remain to be characterized for most TA systems, it has been proposed that when cells are facing environmental stresses, the less stable antitoxin is degraded, leading to an increased production of toxin, resulting in growth arrest of the bacterial cells [5,6]. TA systems are classified into six families, depending on the nature of the antitoxin, i.e., ribonucleic acid (RNA) or protein, and its mechanism of toxin neutralization. For instance, in the most studied family of type II TA systems, the antitoxin is a protein that neutralizes the toxin through direct protein-protein interaction [2,4]. The *Mtb* genome harbors more than 80 TA systems, which were proposed to contribute to virulence and persistence in the infected host [7,8,9,10,11]. Among those, the recently characterized MbcTA system encodes an NAD^+^ phosphorylase toxin, able to trigger bacterial cells death through thorough depletion of the NAD^+^ pool [1]. In the present study, we investigated the expression and regulation of the *mbcAT* operon. We developed a dual fluorescent reporter system, which we used to dissect the operon promoter/operator region at the genetic level. Using this system, we demonstrate that transcription from P*_mbcA_* is induced by a range of stress conditions, reflecting those encountered inside the infected host.

## 2. Results

### 2.1. MbcT Toxin Depletes NAD^+^ and Is Bactericidal in Mycobacterium smegmatis

In order to express the MbcT toxin in *M. smegmatis*, we used an expression vector, pGMCS-TetR-P1-*mbcT*, that can stably integrate into the mycobacterial chromosome, within the *glyV* tRNA gene, through site-specific recombination catalyzed by the L5 mycobacteriophage integrase [12]. In this plasmid, *mbcT* is expressed under the tetracycline-inducible promoter P_myc1 tetO_ [13], hereafter abbreviated P1. Upon transformation into a *Mtb* strain deleted for the endogenous *mbcAT* operon, pGMCS-TetR-P1-*mbcT* conferred anhydro-tetracycline (Atc)-dependent depletion of the essential metabolite NAD^+^, resulting in bacterial cell death [1]. In contrast, growth of a derivative of wild-type *M. smegmatis* strain mc^2^ 155 carrying this plasmid was not affected by addition of Atc to its growth medium (Figure 1a), presumably because the amount of MbcT produced in such conditions was not sufficient to deplete NAD^+^ from the *M. smegmatis* cells (Figure 1d). To verify this hypothesis, we constructed plasmids expressing *mbcT* under the control of stronger Atc-inducible promoters, namely P606Pld and P606 (obtained from D. Schnappinger, Weill Cornell Medical College, NY, USA). Induction of P606Pld-*mbcT* had no effect on mc^2^ 155 growth (Figure 1b), although it reduced the NAD^+^ content to 8% of the amount measured in the uninduced strain carrying pGMCS-TetR-P1-*mbcT*. In contrast, *mbcT* expression from the Atc-induced P606 promoter abolished mycobacterial growth (Figure 1c) and, accordingly, reduced the amount of NAD^+^ to less than 2%, compared to that in control cells (Figure 1d). Furthermore, it resulted in an approximately 99% loss of colony forming units (CFU) within few hours following induction (Figure 1e), and assessment of cell viability using the LIVE/DEAD BacLight stains followed by flow cytometry analysis demonstrated that the proportion of propidium iodide-permeable cells, which was very low in uninduced exponentially growing cells (Figure 1e), increased markedly by 8 or 24 h after the addition of Atc (Figure 1f–i). This confirmed the bactericidal effect of the NAD^+^ depletion following the induction of P606-*mbcT*. In conclusion, providing that it is driven by a relatively strong promoter, production of MbcT in *M. smegmatis* can recapitulate the bactericidal effect of the toxin previously observed in *Mtb*.

### 2.2. Construction of a Dual Fluorescent Reporter System to Assess Gene Expression in Mycobacteria

In order to monitor the expression of mycobacterial genes of interest in in vitro cultures or during infection, we developed a dual fluorescent reporter system in which an integrative plasmid carries both (i) a constitutive reporter, serving as an internal standard or probe for the detection of the bacterial cells and (ii) an additional reporter, expressed from the promoter/operator region of the gene under investigation. This reporter system was constructed by multisite gateway recombination [14], using plasmid pDE43-MCS as backbone, and entry vectors carrying either a 5*′*, a middle or a 3*′* module, as depicted in Figure 2a. The 5*′* module consists of a fluorescent reporter, which can be either green fluorescent protein (GFP), mCherry, mTurquoise, or mVenus, expressed from the constitutive P1 promoter. The middle entry clone introduces a second fluorescent reporter gene, from the same list but different from the first one, fused to the promoter/operator region of the *mbcAT* operon. Finally, the 3*′* entry clone introduces either nothing, using an empty entry clone (pEN23A-MluI), or any gene(s) of interest, which effect on P*_mbcA_* is to be tested. Figure 2b depicts an example of such construct, pGMCS-P1-mCherry-P*_mbcA_*-GFP-*mbcA*, in which the *mbcA* gene is inserted to assess the effect of MbcA on its own transcription.

### 2.3. Use of a Dual Fluorescent Reporter System to Study the Expression of mbcAT

#### 2.3.1. Auto-Repression of *mbcAT*

A set of three plasmids, namely pGMCS-P1-mCherry-P*_mbcA_*-GFP, pGMCS-P1-mCherry-P*_mbcA_*-GFP-*mbcA*, and pGMCS-P1-mCherry-P*_mbcA_*-GFP-*mbcAT*, was constructed and transformed into *M. smegmatis* mc^2^ 155. The resulting strains were grown until exponential phase in complete 7H9 medium and the fluorescence of mCherry and GFP were measured by flow cytometry. The untransformed mc^2^ 155 strain exhibited a very low background level of fluorescence (Figure 3a). In contrast, the three transformed strains exhibited both red and green fluorescence (Figure 3b–d). Comparison of the mean fluorescence intensities showed a similar level of fluorescence for mCherry in the three constructs, whereas the fluorescence for GFP was approximately 5-fold lower when the MbcA antitoxin was expressed, regardless of MbcT toxin co-expression (Figure 3h). Therefore, we conclude that the transcription from P*_mbcA_* is auto-repressed by MbcA, independently of the presence of the toxin. We confirmed these observations in a qualitative assay, by observation of the bacterial cells in fluorescence microscopy (Appendix A). In addition, we also introduced the same set of plasmids into *M. tuberculosis* H37Rv and observed the cells in fluorescence microscopy after fixation with *p*-formaldehyde. As expected, in the wild-type H37Rv, expression of GFP remained very low in all our constructs because of the MbcA repressor produced from the chromosomal copy of the *mbcAT* operon. In contrast, in H37Rv ∆*mbcAT*::Kan^R^, the pattern of mCherry and GFP expression was comparable to that observed in *M. smegmatis*, demonstrating that autoregulation occurs similarly in the two mycobacterial species.

Analysis of the MbcA sequence predicts a helix-turn-helix motif between amino-acids 20 and 40, forming a putative DNA binding domain. In order to test whether this motif is important for repression by MbcA, the Arg^33^ and Arg^37^ residues were substituted with glutamic acid by site-directed mutagenesis of pGMCS-P1-mCherry-P*_mbcA_*-GFP-*mbcA*, and the resulting plasmids were transformed into *M. smegmatis* mc^2^ 155. Both modifications abolished the repression by the MbcA variants, leading to an expression level comparable to that observed in the absence of MbcA (Figure 3h). When the R^33^E substitution was introduced in the plasmid pGMCS-P1-mCherry-P*_mbcA_*-GFP-*mbcAT*, the co-expression of the toxin MbcT with the R^33^E variant of MbcA still resulted in absence of repression.

#### 2.3.2. Genetic Dissection of the *mbcAT* Promoter/Operator Region

In the plasmids described above (Figure 3), the *mbcAT* promoter is located on a 261-bp fragment (P*_mbcA_*_(261)_-GFP, labeled in red letters in Figure 4a) amplified with the oligonucleotides Fw1/Rv1 (Appendix A). A shorter 103-bp fragment, extending between oligonucleotide Fw2 and Rv1, also contains the entire promoter and operator, because it produced the same pattern of expression (P*_mbcA_*_(103)_-GFP, labeled in green letters in Figure 4b). A putative -10 element of *σ*^A^-dependent promoters is present upstream from the *mbcA* transcription start site identified by RNA-seq experiments [15]. We confirmed this promoter location by generating two 239-bp DNA fragments amplified with the oligonucleotides Fw1/Rv2, with the wild-type sequence and Fw1/Rv3, with a mutation changing a consensus T into a non-consensus G in the -10 element. Whereas the first fragment still allowed the expression of GFP (P*_mbcA_*_(239)_-GFP, labeled in blue letters in Figure 4b), the T=>G substitution almost completely abolished the production of the downstream encoded GFP (P*_mbcA_*_(239-G)_-GFP, labeled in brown letters in Figure 4b). In addition, we observed that the Fw1/Rv2 fragment was much less sensitive to repression by MbcA. To further analyze the *mbcA* operator, we constructed mutations of 4 (Mut1; changing CAAA into TCGT) or 3 nucleotides (Mut2; changing ACA into GTG), and we observed that Mut1 resulted in a high expression in the presence of MbcA, whereas Mut 2 did not abolish repression by MbcA.

### 2.4. Multistress Induction of mbcAT Expression

To assess the regulation of *mbcAT* expression, the plasmid pGMCS-P1-mTurquoise-P_mbcA_-mVenus-*mbcA* was introduced into wild-type Mtb H37Rv, and the resulting strain was grown to mid-exponential phase and subjected to various stress conditions. As shown in Figure 5, we could observe qualitatively that the expression from P*_mbcA_* was increased upon starvation following incubation of the cells in phosphate saline buffer (PBS), or upon treatment with H_2_O_2_ or with the NO-generating reagent diethylenetriamine/nitric oxide adduct (DETA/NO).

Finally, *Mtb* strain H37Rv/pGMCS-P1-mTurquoise-P*_mbcA_*-mVenus-*mbcA* was used to infect mouse or human macrophages, and fluorescence microscopy of fixed infected macrophages demonstrated that transcription from P*_mbcA_* is stimulated upon *Mtb* phagocytosis by macrophages (Figure 6).

## 3. Discussion

The dual fluorescent reporter system presented here, constructed using the Gateway cloning technology [12], allows for quantification and/or visualization of bacterial cells, and in parallel for following expression of a gene of interest (Figure 2). As shown here, this system is stable and robust, since it is based on an integrative vector and exhibited only minor variations of the expression of the internal standard reporter (mCherry in Figure 3 and Figure 4). Allowing both quantitative (Figure 3 and Figure 4) and qualitative measurements (Figure 5 and Figure 6), this system can be used to follow gene expression not only in bacterial cultures, or inside infected macrophages, as shown in this work, but also inside host tissues during infection of animal models [16]. Finally, it is also very versatile, since it offers a straightforward way to construct bacterial variants carrying a set of different reporters, as exemplified here with a variety of fluorescent proteins (Figure 2). We believe that it can be of broad interest to investigate gene expression and regulation in mycobacteria.

RNA-seq experiments indicated a potential transcription start site (TSS) of *mbcA* in *M. tuberculosis* (indicated as +1 in Figure 4) and located 34 nucleotides upstream from its translation initiation site (TIS) [15]. Based on bioinformatics analysis, it has been proposed that transcription of the *mbcAT* operon occurs from two overlapping promoters recognized by RNA polymerase holoenzyme using alternative sigma factors [17]. An upstream promoter, with a *σ*^L^-dependent -10 element (GGTTC) located at position -54 to -50 from the TIS, and a second promoter, using a *σ*^H^-dependent -10 element (CGTGTC), located at positions -50 to -45. The *σ*^L^ and *σ*^H^ alternative sigma factors are both present in *M. smegmatis* [18] and might therefore be involved in *mbcAT* expression in both mycobacterial species used in this study. Our data do not rule out the possibility of *mbcAT* expression from *σ*^L^- or *σ*^H^-dependent promoters, but they demonstrate that yet another -10 element is essential for the expression of *mbcAT* (Figure 4), which sequence **TG**TC**A**TAA**T** fits (nucleotides in bold letters) the consensus of *σ*^A^-dependent promoters with an extended -10 motif (TGNTANNNT [15]). The involvement of this -10 element in *mbcAT* expression has been confirmed by our genetic analysis since its activity was abolished by a mutation changing a consensus base of the -10 element into a non-consensus one, at position -40 with respect to the TIS (Figure 4).

Many type II TA systems are regulated by auto-repression, with the antitoxin binding to an operator site in the vicinity of the operon promoter [2,4]. It is also the case for the *mbcAT* operon (Figure 3 and Figure 4). In solution, MbcA and MbcT form a heterododecameric complex (3 × [MbcT-MbcA]_2_) arranged around a 3-fold symmetry axis in which MbcA folds into a single structured domain consisting of seven α helices [1]. Helices α2 and α3 form a putative helix-turn-helix motif, and Arg^33^ and Arg^37^, which are pointing out of helix 3, could be important for the recognition of the operator site. Consistently, substitutions of any of these two arginine residues totally abolished the transcription repression by MbcA (Figure 3). The *mbcA* promoter region harbors a direct repeat of two GACAAA motifs, separated by 12 bp (underlined with a magenta line in Figure 4a). Deletion of the downstream motif on promoter fragment Fw1-Rv2 abolishes the repression by MbcA and mutation Mut1, which disrupts this downstream motif, increases transcription in the presence of MbcA to the level observed in absence of the repressor. In contrast, modification of nucleotides downstream from this motif (mutation Mut2) did not relieve the repression by MbcA (Figure 4b). Although additional biochemical experiments are needed to determine the precise interaction of MbcA with its operator site, these observations suggest that MbcA could bind the GACAAA direct repeats and that both repeats are indispensable to constitute a functional operator.

Several autoregulated type II TA systems are subject to the so-called conditional cooperativity, i.e., a repression mechanism in which the antitoxin-repressor can participate in different types of complexes, alone, or with its cognate toxin, exhibiting different repression efficacies [2,4,19]. Conditional cooperativity has been proposed to contribute to several aspects of the TA systems biology [2]. In such systems, the antitoxin alone binds its operator site with low affinity, resulting in poor repression. When toxin and antitoxin are present in stoichiometric amounts, binding of the toxin to the antitoxin increases its affinity for the operator, leading to a more potent repression of transcription than is observed for the antitoxin alone. Finally, high amounts of toxin can result in a different heteromeric complex exhibiting low affinity for the operator, thereby relieving repression in excess of toxin. As already shown for a number of TA systems [20], our data suggest that the MbcTA system is not subject to conditional cooperativity, since the level of transcription repression by the antitoxin appeared independent on the production of the toxin (Figure 3). We note that in the dodecameric MbcTA complex, the helix-turn-helix motif of MbcA, necessary for the repression (Figure 3), is masked by an interaction with the α6 helix of MbcT [1]. In consequence, the dodecameric complex is most likely unable to participate in repression. In the wild-type H37Rv *M. tuberculosis* strain, the ectopic expression of MbcT under the control of the P1 promoter revealed not to be toxic [1], indicating that expression of the wild-type *mbcAT* operon leads to higher amounts of MbcA antitoxin than of MbcT toxin. Therefore, only a fraction of MbcA must be is involved in MbcT-neutralizing complexes, and it is likely that the remaining excess of MbcA can function as auto-repressor.

Previous work has already reported increased expression of *mbcAT* in stress conditions. The MbcT toxin (Rv1989c) has been identified by proteomic analysis in culture filtrates of PBS-starved *M. tuberculosis* cells, whereas it is not detectable during exponential growth [21]. Transcriptomic studies demonstrated increased amounts of *mbcA* and/or *mbcT* mRNA in stationary compared to exponential phase [15], during the so-called enduring hypoxic response [22], or in *M. tuberculosis* persisters surviving a treatment with D-cycloserine [8]. Our data are in agreement with these observations, and further demonstrate that transcription of *mbcAT* increases upon exposure to oxidative stress caused by H_2_0_2_ or NO-generating compounds (Figure 5), conditions mimicking those encountered inside macrophages [23,24,25]. In agreement, we also observed increased transcription from the *mbcAT* promoter within infected macrophages (Figure 6). To date, the physiological role of the MbcTA system has not been firmly established. However, the existence of a gene mimicking the antitoxin MbcA in the genome of several myco-bacteriophages [1] strongly suggests that this system could belong to the family of TA systems used to fight against phage infections [26,27]. The multi-stress induction of *mbcAT*, leading to accumulation of both the toxin and the antitoxin, should not be toxic for the *M. tuberculosis* cells. However, it is reminiscent of the so-called general stress response observed in many bacterial species, which results in the expression of a variety of adaptive responses to stresses not yet encountered by bacterial cells but that contribute to reinforce their resistance if they eventually happen to deal with one of these stresses [28].

The *M. tuberculosis* genome is particularly rich in TA systems [7,8,9,10,11], and, as already proposed for other bacterial pathogens [29,30], it is tempting to speculate that at least some of these systems could be used as novel tools in adjunct therapies with classical antibiotics. However, a number of studies have proposed that activation of TA systems may trigger a non-growing physiological state, or so-called persistence, in which the bacterial cells become tolerant to antibiotics [31,32,33,34]. Although the role of TA systems in the development of tolerance is still a matter of debate [35], this raises a serious drawback to their potential therapeutic use. The MbcTA system attracted our attention because it is one of the only two TA systems of *M. tuberculosis* that harbor an essential antitoxin gene, i.e., which cannot be genetically disrupted [36]. We reasoned that this could be the consequence of an atypical and particularly toxic activity of MbcT. Indeed, we have demonstrated that MbcT activity, when expressed in recombinant conditions generating physiological expression levels of the toxin, is bactericidal for *M. tuberculosis* through depletion of NAD^+^ [1], a metabolite known to be essential for mycobacteria survival, even in a dormant state [37,38,39]. In addition, we have shown that production of the MbcT toxin during infection could act in synergy with isoniazid treatment to reduce the bacterial load in mice infected by *M tuberculosis*, supporting the idea that MbcTA could be a novel therapeutic target [1]. The observations reported here strengthen this proposal, given that the induction of *mbcAT* during infection ensures the presence of the toxin inside the infecting *M. tuberculosis* cells, where targeting its activation could help fighting against TB.

## 4. Materials and Methods

### 4.1. Bacterial Strains and Cultures

*M. tuberculosis* strains H37Rv (ATCC27294) and H37Rv ∆*mbcAT*::Kan^R^ [1] and *M. smegmatis* wild-type strain mc^2^ 155 (ATCC700084) were routinely grown aerobically at 37 °C in complete 7H9 medium, i.e., Middlebrook 7H9 medium (Difco, Franklin Lakes, NJ, USA) supplemented with 10% albumin-dextrose-catalase (ADC, Difco) and 0.05% Tween 80 (Sigma-Aldrich, Saint-Louis, MI, USA) or on Middlebrook 7H11 agar medium (Difco) supplemented with 10% oleic acid-albumin-dextrose-catalase (OADC, Difco). When required, streptomycin (25 μg mL^−1^), kanamycin (50 μg mL^−1^), or anhydrotetracycline (Atc; 200 ng mL^−1^) were added to the culture media. *E. coli* Stellar strain (Clontech Laboratories, Inc., Mountain View, CA, USA) was grown aerobically at 37 °C in L Broth medium or on L Agar, supplemented with streptomycin (25 μg mL^−1^).

### 4.2. Oligonucleotides and Plasmids Used in This Work

Oligonucleotides and plasmids used in this work are listed in Appendix A, respectively. Plasmids pEN- and pGMCS- were constructed by multisite gateway recombination [14] following the manufacturer’s instructions (Life Technologies, Carlsbad, CA, USA). DNA fragments were amplified by PCR, using Phusion DNA polymerase (Thermo-Fisher Scientific, Waltham, MA, USA) with the templates and oligonucleotide pairs indicated in Appendix A. Plasmids pEN41- and pEN12- were generated by “BP” cloning, using pDO41A- or pDO221A- as destination vectors, and DNA fragments flanked by *attB4*/*attB1* or *attB1*/*attB2*, respectively. Plasmids pGMCS-TetR-P1-*mbcT*, pGMCS-TetR-P606Pld-*mbcT* and pGMCS-TetR-P606-*mbcT* were constructed by “LR” cloning reactions, using plasmid pDE43-MCS as destination vector, plasmid pEN41-TetR as 5′ entry clone, plasmids pEN12-P1, pEN12-P606Pld, or pEN12-P606 as middle entry clone, and plasmid pEN23-*mbcT* as 3′ entry clone. Plasmids pGMCS-P1-reporter1-P*_mbcA_*-Reporter2 were constructed by “LR” cloning reactions, using plasmid pDE43-MCS as destination vector, a plasmid pEN41-P1-Reporter as 5′ entry clone, a plasmids pEN12-P*_mbcA_*-reporter as middle entry clone, and plasmids pEN23-MluI, pEN23-*mbcA*, or pEN23-*mbcAT* as 3′ entry clone. Site directed mutagenesis of plasmids was performed by amplification by PCR of the plasmid to be mutated with a pair of overlapping oligonucleotides carrying the mutation to be introduced (Appendix A). The linear plasmid DNA was purified on agarose gels and circularized using HD In-Fusion cloning reactions (Takara, Kusatsu, Japan). All plasmid constructs were verified by DNA sequencing.

### 4.3. M. Smegmatis Viability Assays and NAD^+^ Measurement

Exponentially growing cultures (OD_600_ between 0.05 and 0.2) of *M. smegmatis* mc^2^ 155 transformed with the desired plasmids were divided in two: half was left in the same medium (uninduced cultures) and the other half was treated with 200 ng mL^−1^ of Atc to induce expression from the tetracycline-inducible promoters. After various time post-induction, samples were harvested and centrifuged to eliminate residual Atc. Cells were resuspended in PBS buffer and dilutions were plated on L Agar supplemented with streptomycin, colonies were counted after 3 days at 37 °C. For labeling with LIVE/DEAD^®^ BacLight dyes (Molecular Probes, Eugene, OR, USA), cells were harvested 8 or 24 h post-Atc induction. Cells were centrifuged, resuspended in PBS buffer and stained as recommended by the manufacturer. Labeled cells were analyzed by fluorescence activated cell sorting using a BD FACS LSRFortessa X20 flow cytometer. Flow cytometry data analysis were performed using FlowJo software (Version 10; Becton, Dickinson and Company, Ashland, OR, USA).

Samples of the above cultures were harvested 8 h post-Atc induction and centrifuged. Cells were resuspended in PBS, adjusting the OD_600_ to 0.2. 0.1 μm-diameter glass beads were added to tubes containing 500 µL of each cell suspensions, and cells were lysed by four 60-s pulses at full speed in a bead-beater device. The samples were centrifuged for 1 min at 20,200× *g* and 50 µL of the lysates were mixed with an equal volume of NAD/NADH-Glo™ Detection Reagent (Promega). Luciferin bioluminescence was measured after 30 min of incubation at room temperature, using a CLARIOstar^®^ plate reader (BMG LABTECH, Champigny s/Marne, France) and normalized to background signal (PBS-only) to monitor the relative amounts of NAD^+^ present in the cell extracts.

### 4.4. Effect of Stress Conditions on P_mbcA_

Exponential cultures in complete 7H9 medium of *M. tuberculosis* H37Rv transformed with plasmid pGMCS-P1-mTurq-P*_mbcA_*-mVenus-*mbcA* were harvested and centrifuged. Cell pellets were and resuspended in PBS buffer (for starvation), or complete 7H9 medium supplemented with 5 mM H_2_O_2_ or 0.5 mM DETA/NO. At different times, samples were harvested, fixed with 4% *p*-formaldehyde and observed by fluorescent microscopy using large field Leica DMIRB, magnification 630×.

### 4.5. Human and Mice Macrophage Cultures and Infection

Human monocytes were obtained from healthy blood donors by Etablissement Français du Sang, EFS, Toulouse, France). Written informed consents were obtained from the donors before sample collection (under EFS Contract n°21PLER2017-0035 valid until July 02 2020, which was approved by the French Ministry of Science and Technology, agreement nr. AC2009-921, following articles L1243-4 and R1243-61 of the French Public Health Code). Monocytes were prepared following a previously published procedure [40]. Cells were purified using CD14 microbead positive selection and MACS separation columns (Miltenyi Biotec, Bergisch Gladbach, Germany), according to manufacturer’s instructions. For differentiation of monocyte-derived macrophages, monocytes were allowed to adhere to glass coverslips (VWR international, Radnor, PA, USA) in 6-well plates (Thermo-Fisher Scientific), at 1.5 × 10^6^ cells/well, for 1 h at 37 °C in pre-warmed RPMI-1640 medium (GIBCO, Thermo-Fisher Scientific, Waltham, MA, USA). The medium was then supplemented with the following additives: 10% Fetal Bovine Serum (Sigma-Aldrich), 1 mM sodium pyruvate (GIBCO), 50 µM β-mercaptoethanol (GIBCO), and 20 ng mL^−1^ human Macrophage Colony-Stimulating Factor (Miltenyi Biotec, Bergisch Gladbach, Germany). Cells were allowed to differentiate for seven days at 37 °C under 5% CO_2_ atmosphere. Murine bone-marrow derived macrophages were prepared as described [41]. Bone-marrow cells were flushed out of the femurs and tibias of 6- to 8-week-old female C57BL/6 mice and cultured in Petri dishes (2 × 10^6^ cells per dish) in pre-warmed RPMI-1640 GlutaMax medium (GIBCO or Life Technologies, Carlsbad, CA, USA), supplemented with 10% FBS (Sigma-Aldrich), 1 mM sodium pyruvate (GIBCO or Life Technologies), 50 µM β-mercaptoethanol (GIBCO or Life Technologies), and 20 ng mL^−1^ M-CSF (PeproTech, Rocky Hill, NJ, USA) at 37 °C under 5% CO_2_ atmosphere. Medium was refreshed every three days of culture.

For infection, *M. tuberculosis* strain H37Rv/pGMCS-P1-mTurq-P*_mbcA_*-mVenus-*mbcA* was grown to exponential phase in complete 7H9 medium. Mycobacterial clumps were disaggregated by at least 20 passages through a 25G needle, and macrophages were infected at MOI of 0.1 in complete RPMI medium for 4 h at 37 °C under 5% CO_2_ atmosphere. Cells were then washed with RPMI and further incubated at 37 °C in RPMI supplemented medium. Infected cells were harvested at day 0 or 24 h post-infection, fixed with 4% *p*-formaldehyde and observed by fluorescent microscopy using large field Leica DMIRB, at a magnification of 630X.

## Figures and Tables

**Figure 1 toxins-12-00329-f001:**
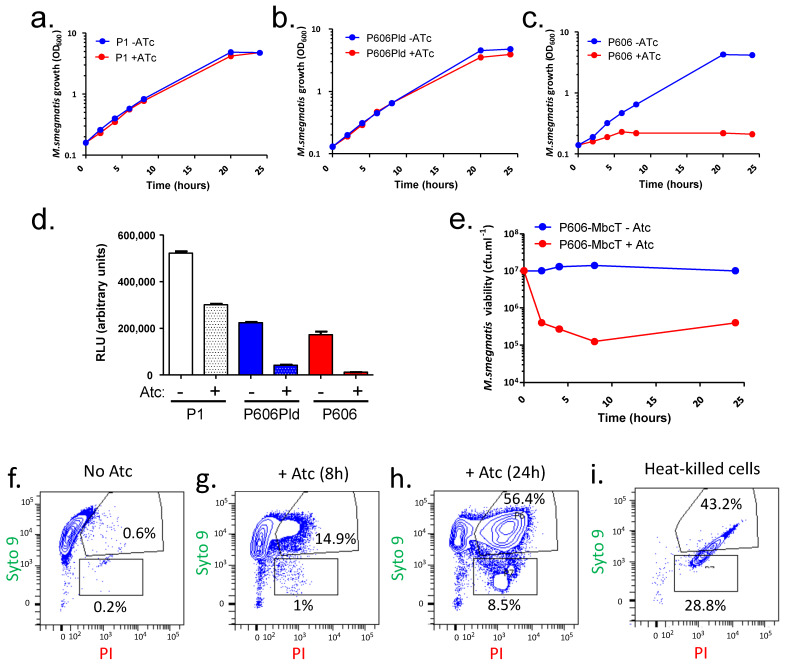
The toxin MbcT depletes NAD^+^ and is bactericidal in *Mycobacterium smegmatis*. (**a**–**c**) Cultures of *M. smegmatis* strain mc^2^ 155 transformed with plasmids producing MbcT under the control of the indicated promoter were diluted at time 0 in fresh medium supplemented with streptomycin alone (blue) or streptomycin and 200 ng.mL^−1^ anhydro-tetracycline (Atc) (red) and bacterial growth was followed over up to 24 h. (**d**) Samples of the *M. smegmatis* cultures were harvested after 8 h of treatment and relative content of NAD^+^ was measured in cell extracts. Values are representative of two independent replicates of the same experiment. (**e**) Samples of the cultures of *M. smegmatis* mc^2^ 155/pGMCS-TetR-P606-mbcT were harvested at the indicated times, diluted, and plated on L agar. Colonies were counted after 3 days of growth at 37 °C. (**f**–**h**) Samples of the cultures of mc^2^ 155/pGMCS-TetR-P606-mbcT were harvested after 8 or 24 h of treatment with Atc, labeled with the LIVE/DEAD BacLight dyes (Syto9 and Propidium Iodide (PI)) and analyzed by flow cytometry (FACS). (**i**) Cells grown without Atc were heat-killed at 98 °C for 30 min before LIVE/DEAD labeling and FACS analysis. Data are representative of two experiments with similar results.

**Figure 2 toxins-12-00329-f002:**
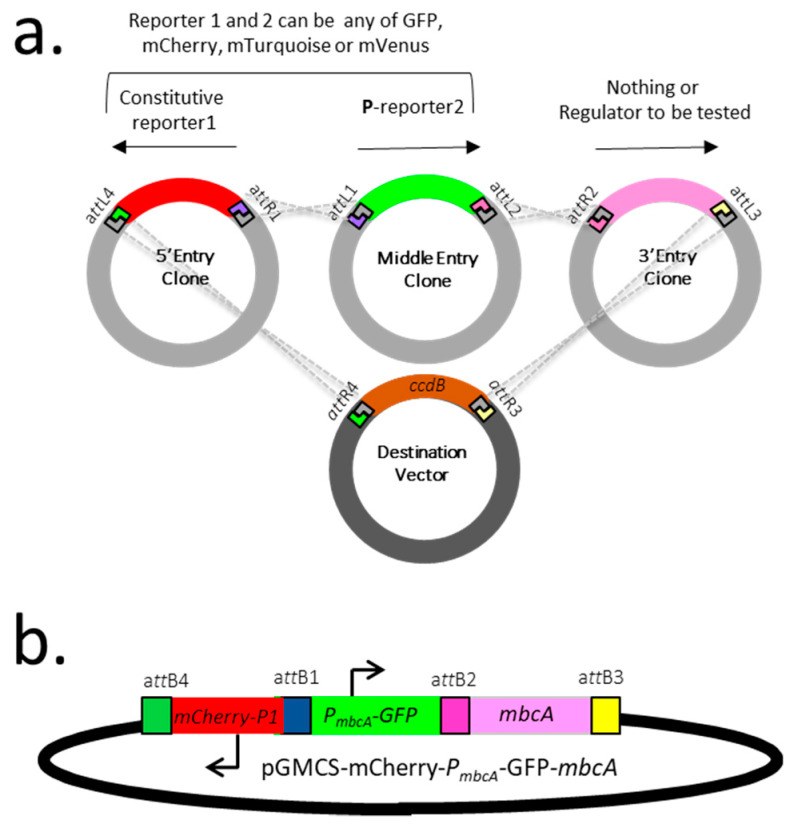
A dual florescent protein reporter system developed for studying gene expression in mycobacteria. (**a**). Gateway cloning strategy used to construct the required plasmids. (**b**). Example of plasmid construction designed to analyze the expression and regulation of *mbcAT*. GFP = green fluorescent protein.

**Figure 3 toxins-12-00329-f003:**
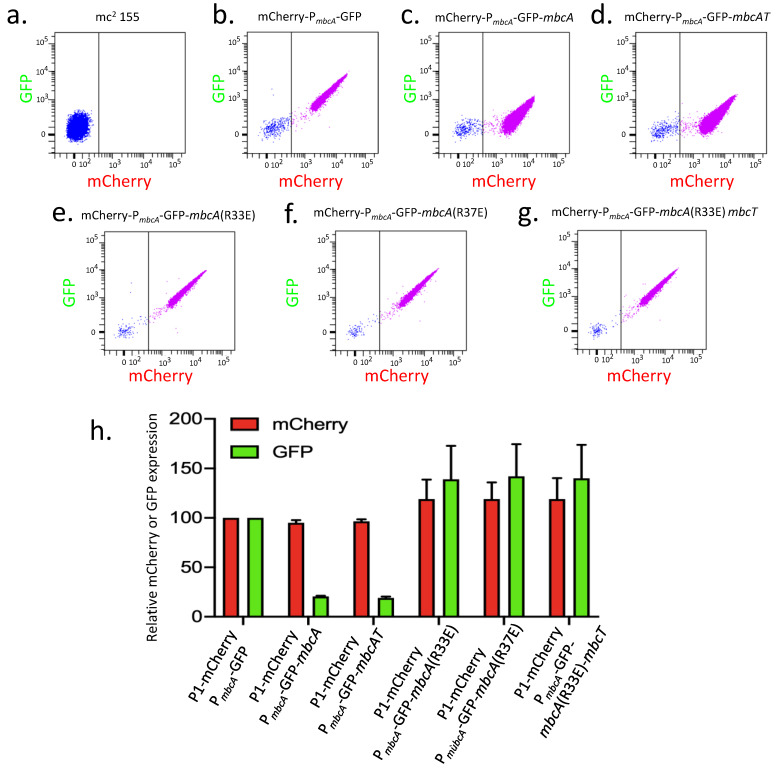
Autorepression of the *mbcAT* operon. (**a**)–(**g**) *M. smegmatis* mc^2^ 155 wild-type and transformed with the indicated pGMCS-mCherry-P*_mbcA_*-GFP derivatives were grown to exponential phase in complete 7H9 medium and analyzed by flow cytometry. Blue and pink points separated by a vertical line indicate the cells negative or positive for mCherry fluorescence, respectively. (**h**) Mean fluorescence intensities (MFI) of the fluorescence positive cells (pink points) relative to that of the strain harboring pGMCS-mCherry-P*_mbcA_*-GFP (a) are reported for mCherry (red bars) and for GFP (green bars). Values are the average of two biological replicates with standard deviation.

**Figure 4 toxins-12-00329-f004:**
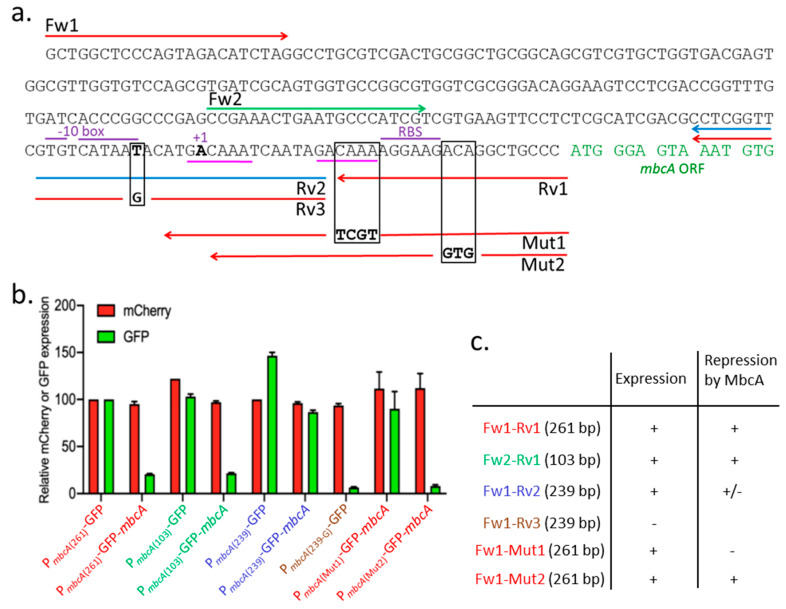
Genetic analysis of the promoter/operator region of the *mbcAT* operon. (**a**) DNA sequence of the *mbcA* promoter/operator region. Arrows indicate oligonucleotides used to amplify various fragments encompassing the promoter region. Black squares surround the nucleotides modified by mutations introduced in the oligonucleotides Rv3 (T=>G), Mut1 (CAAA=>TCGT), and Mut2 (ACA=>GTG). +1 points to the transcription start site. (**b**) Mean fluorescence intensities (MFI) were determined by flow cytometry with strains harboring the different constructs. Red and green bars indicate MFI relative to that of strain harboring pGMCS-mCherry-P_MbcA_-GFP. Values are the average of two biological replicates with standard deviation. (**c**) Summary of the results obtained with the different constructs.

**Figure 5 toxins-12-00329-f005:**
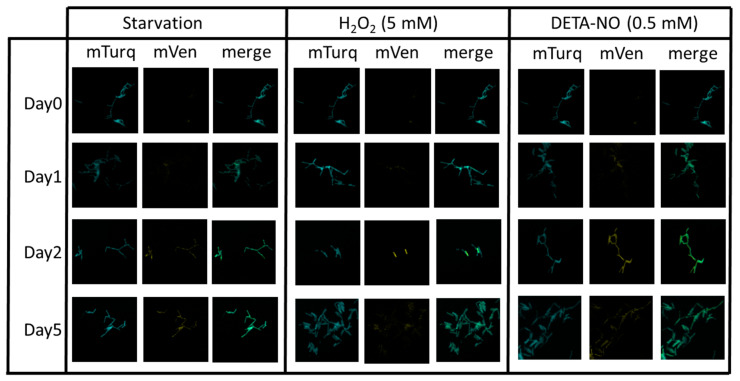
Effect of stress conditions on expression of the *mbcAT* operon in *M. tuberculosis.* Wild-type H37Rv transformed with plasmid pGMCS-P1-mTurq-P*_mbcA_*-mVenus-*mbcA* was grown in complete 7H9 medium. At day 0, cells were harvested and resuspended in phosphate saline buffer (PBS) buffer (starvation), or 7H9 medium supplemented with 5 mM H_2_O_2_ or 0.5 mM DETA/NO. At the indicated times, samples were harvested, fixed with 4% *p*-formaldehyde and observed by fluorescent microscopy using large field Leica DMIRB, magnification 630×.

**Figure 6 toxins-12-00329-f006:**
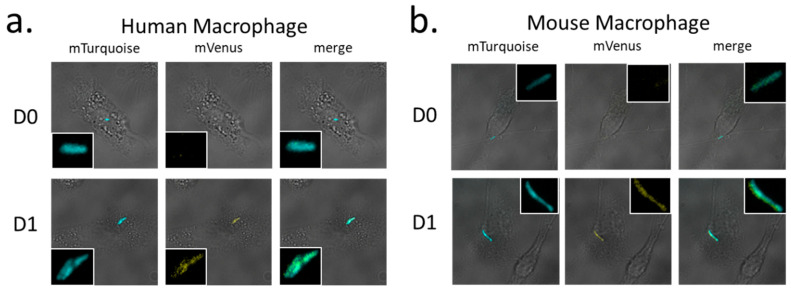
Induction of the *mbcAT* operon expression in *M. tuberculosis* upon infection of macrophages. *Mtb* H37Rv/pGMCS-P1-mTurq-P*_mbcA_*-mVenus-*mbcA* was used to infect human (**a**) or mouse (**b**) macrophages at multiplicity of infection of 0.1. Infected cells were harvested at day 0 (D0) or 24 h post-infection (D1), fixed with 4% *p*-formaldehyde and observed by fluorescent microscopy using large field Leica DMIRB, magnification 630×. Images show the merged bright field and either the blue, yellow, or merged fluorescence. Insets show enlarged view of the phagocytized bacteria.

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
