# Peer review of "Multi-Stress Induction of the *Mycobacterium tuberculosis* MbcTA Bactericidal Toxin-Antitoxin System"

_toxins, 2020, doi:10.3390/toxins12050329_

Round 1

Reviewer 1 Report

In this manuscript the authors study the role and expression of the MbcTA operon in details. They used a dual reported system combining a internal standard and the reporter to be studied. This manuscript is very well presented and organized, the methodology is adequate.

I recommend this manuscript for publication.

One typo is found in the legend of Figure 1: "treatemant" and "(h)" should read "(i)". The (i) figure title seems to be cut.

Author Response

Manuscript ID: toxins-804185

Title: Multi-stress induction of the Mycobacterium tuberculosis MbcTA bactericidal toxin-antitoxin system.

Response to Reviewers

We would like to thank the Reviewers for their positive comments and constructive criticism that will improve the clarity of the manuscipt. Point-by-point responses to Reviewers comments are listed below in blue, and where necessary, changes have been made to the peer review version of the manuscript and are shown in blue highlight. In some instances, changes have been made to figures but these are not altered in color, in order to maintain consistency within the figure.

Reviewer #1

In this manuscript the authors study the role and expression of the MbcTA operon in details. They used a dual reported system combining a internal standard and the reporter to be studied. This manuscript is very well presented and organized, the methodology is adequate.

I recommend this manuscript for publication.

One typo is found in the legend of Figure 1: "treatemant" and "(h)" should read "(i)". The (i) figure title seems to be cut.

We thank the Reviewer for his/her positive evaluation of our manuscript.

Both typos were corrected in the legend.

Another typo appeared during the transformation of our original version into the peer review version: Mycobacterium smegmatis and M. smegmatis are names of bacterial species and must be in italic letters.

Reviewer 2 Report

This manuscript describes the construction and analysis of a dual reporter system for the following the expression of the M. tuberculosis operon encoding the McbTA type II toxin-antitoxin module both in culture and in mice. The paper is well-written and tidy. Although the conclusions about autoregulation are not dramatically different from other descriptions of TA systems (in this case the antitoxin is sufficient for full repression) the description of the reporter system will interest readers. My main issue is related to quantification and statistical analysis pointed out below. Otherwise, I have only a few minor suggestions

Fig. 3: It will not be immediately obvious to the average reader how to interpret the data in panels a-g. This should be described in more detail in the legend. Also, what is blue vs pink color; what does vertical line signify and P6? Shouldn't the y-axis label be GFP?

Error bars should be added to the histograms in panel h

L155 and L218 The data for the T->G substitution does not appear to be shown

L156 Describe the Mut1 and Mut2 mutations in the text

Fig. 4 Add error bars in panel b 

Fig. 5 Some quantification and statistics are needed for this figure

L230 Describe the non-effect of the mut2 mutation in the text

L251 I disagree with this statement ; the overlapping reading frames and translational coupling is more likely to ensure stoichiometric quantities of each, than more antitoxin

L263 What does increased expression of whole operon mean for bacterial physiology? If both T and A are expressed, then no there should be no difference in toxicity in these conditions? Are protease levels likely to be increased under these conditions to degrade the antitoxin ? Please include this point in discussion.

Author Response

Manuscript ID: toxins-804185

Title: Multi-stress induction of the Mycobacterium tuberculosis MbcTA bactericidal toxin-antitoxin system.

Response to Reviewers

We would like to thank the Reviewers for their positive comments and constructive criticism that will improve the clarity of the manuscipt. Point-by-point responses to Reviewers comments are listed below in blue, and where necessary, changes have been made to the peer review version of the manuscript and are shown in blue highlight. In some instances, changes have been made to figures but these are not altered in color, in order to maintain consistency within the figure.

Reviewer #2

This manuscript describes the construction and analysis of a dual reporter system for the following the expression of the M. tuberculosis operon encoding the McbTA type II toxin-antitoxin module both in culture and in mice. The paper is well-written and tidy. Although the conclusions about autoregulation are not dramatically different from other descriptions of TA systems (in this case the antitoxin is sufficient for full repression) the description of the reporter system will interest readers. My main issue is related to quantification and statistical analysis pointed out below. Otherwise, I have only a few minor suggestions.

We thank the Reviewer for his/her positive evaluation of our manuscript.

  • 3: It will not be immediately obvious to the average reader how to interpret the data in panels a-g. This should be described in more detail in the legend. Also, what is blue vs pink color; what does vertical line signify and P6? Shouldn't the y-axis label be GFP? Error bars should be added to the histograms in panel h

We modified the legend of Fig. 3 to clarify the figure.

The “RFP” typo was corrected into “GFP”. “P6” that corresponded to the mCherry positive fraction of the cells in the FACS gating strategy has been removed.

A new panel h including error bars has been included (with accompanying modification in the legend).

  • L155 and L218 The data for the T->G substitution does not appear to be shown

These data are shown in Figure 4. In this figure, we used a color code to indicate which fragments were used for each genetic construct and to improve clarity, we now mention this color code in the text (L147, 150, 155 and 157).

We specifically modified L156-157: “production of the downstream encoded GFP (PmbcA(239-G)-GFP, labelled in brown letters in Figure 4b).”

  • L156 Describe the Mut1 and Mut2 mutations in the text

The description has been added (L 159-160).

“To further analyze the mbcA operator we constructed mutations of 4 (Mut1; changing CAAA into TCGT) or 3 nucleotides (Mut2; changing ACA into GTG) and we observed that Mut1 resulted in a high expression in the presence of MbcA, whereas Mut 2 did not abolish repression by MbcA.”

  • 4 Add error bars in panel b 

A new panel b including error bars has been included (with accompanying modification in the legend).

  • 5 Some quantification and statistics are needed for this figure

We fully agree with the Reviewer that quantification and statistical analysis of the images would improve the impact of these data. However, to achieve this goal, we would need to perform new experiments and take additional pictures. Due to the very slow return to normal after the 2-month full close down of our laboratory following the COVID-19 pandemic and the fact that they these experiments are performed with the slow growing bacterium M. tuberculosis in BSL3 conditions, it will not be possible to conduct these experiments before at best the beginning, and most probably rather the end of July.

We have been very careful with the presentation of these data, and stated in the results (L175) and discussion (L202) sections that they are “qualitative” only. We hope that this can be at the Reviewer’s satisfaction.

  • L230 Describe the non-effect of the mut2 mutation in the text

A sentence describing this has been added (L234-235)

“In contrast, modification of nucleotides downstream from this motif (mutation Mut2) did not relieve the repression by MbcA (Figure 4b).”

  • L251 I disagree with this statement; the overlapping reading frames and translational coupling is more likely to ensure stoichiometric quantities of each, than more antitoxin

The paragraph mentioning translational coupling has been removed. We now refer to previously published data indicating that wild type Mtb contains a higher amount of MbcA than of MbcT, supporting the proposed conclusion (L 253-257).

“In the wild type H37Rv M. tuberculosis strain the ectopic expression of MbcT under the control of the P1 promoter revealed not to be toxic [1], indicating that expression of the wild type mbcAT operon leads to higher amounts of MbcA antitoxin than of MbcT toxin. Therefore, only a fraction of MbcA must be is involved in MbcT-neutralizing complexes and it is likely that the remaining excess of MbcA can function as auto-repressor.”

  • L263 What does increased expression of whole operon mean for bacterial physiology? If both T and A are expressed, then no there should be no difference in toxicity in these conditions? Are protease levels likely to be increased under these conditions to degrade the antitoxin ? Please include this point in discussion.

We introduced a new paragraph (L268-276) to discuss the possible role of the multi-stress induction of the mbcAT operon (3 additional references were introduced with this paragraph).

“To date, the physiological role of the MbcTA system has not been firmly established. However, the existence of a gene mimicking the antitoxin MbcA in the genome of several myco-bacteriophages [1] strongly suggests that this system could belong to the family of TA systems used to fight against phage infections [26, 27]. The multi-stress induction of mbcTA, leading to accumulation of both the toxin and the antitoxin, should not be toxic for the M. tuberculosis cells. However, it is reminiscent of the so-called general stress response observed in many bacterial species, that results in the expression of a variety of adaptive responses to stresses not yet encountered by bacterial cells, but that contribute to reinforce their resistance if they eventually happen to deal with one of these stresses [28]“
